# Towards Sustainable Wood-Based Energy: Evaluation and Strategies for Mainstreaming Sustainability in the Sector

**Julia Szulecka** 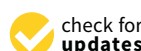

Centre for Technology, Innovation and Culture (TIK), University of Oslo, P.O. Box 1108 Blindern,
NO-0317 Oslo, Norway; julia.szulecka@tik.uio.no

**Abstract:** Bioenergy, mostly from wood biomass, is now widely seen as an important element in the efforts to tame dangerous climate change. At the same time, foresters and development specialists note that wood-based energy production can contribute to rural development. However, to deliver on these two goals without generating negative side effects, wood-based energy has to be sustainable, while currently, the sector is developing rapidly in ways that are technologically advanced, with questionable sustainability. How can sustainability be achieved in bioenergy production, to make it a viable element of climate change mitigation, adaptation, and rural development? Arguing for the need to mainstream sustainability thinking into wood-based energy production, the article draws on a critical literature review to identify four different levels of sustainability in the existing research on bioenergy from wood. It shows two possible strategies for integrating sustainability in wood bioenergy production. A top-down approach draws on global forestry governance instruments, while a bottom-up approach uses best-practices in forest plantations for bioenergy purposes, as illustrated by a case study from rural Paraguay. Using aggregated and visualized sustainability indicators, the article exemplifies what sustainable bioenergy production means in more tangible terms.

**Keywords:** sustainability; bioenergy; governance; fuelwood; certification; out-grower schemes; climate change; rural development; Multi-Criteria Analysis

## 1. Introduction

Faced with the complex challenges of combating climate change and achieving development globally, some policymakers see bioenergy as an important element of future energy mixes [1]. Following the Paris Agreement, most scenarios for keeping global warming under the 2-degree target have included a large share of BECCS—bioenergy combined with carbon capture and storage [2,3]. The development of wood-based bioenergy, from solid biomass, can also bring positive impacts on rural economies in terms of employment, entrepreneurial opportunities and incomes, to a greater extent than any other renewable or non-renewable energy options [4]. These co-benefits are recognized in the developing world, but also in Europe, where the European Union's (EU) 2016 "Clean Energy Package" explicitly emphasizes the role for bioenergy among other renewables [5]. However, it also signals some potential problems that an increased use of solid biomass (wood, but also energy crops like rapeseed, sunflower, and soybean) for energy can bring [6].

Is bioenergy from wood by definition renewable and sustainable? A sustainability index based on 27 parameters shows that biomass (identified as both forest and agricultural residues and energy crops) scores the lowest among "renewables"—0.39, not far from oil and coal, and surpassed by non-renewable nuclear (0.40) and natural gas (0.57) as well as renewable small hydro (0.61), photovoltaic (0.69), onshore wind (0.76) and solar thermal energy (0.80) [7].

While individual indicators can be disputed, the point is clear—bioenergy from wood biomass is not unproblematic. Although renewable, it has significant environmental and social externalities. The UN Food and Agriculture Organization (FAO), underlines that if bioenergy from wood is to be part of our energy future, it needs to see its sustainability safeguarded, which is largely dependent on "the effectiveness of policies and the consistency with which they are implemented" [8]. The recent United Nations flagship initiative, Sustainable Energy for All (SE4All), also recognizes that "sustainably and efficiently produced biomass energy should be central to energy access strategies in poorer countries" [9,10]. Similarly, various driving factors, such as climate change, and the security of energy supplies and rural development, led the International Energy Agency to propose a Bioenergy Agreement (IEA Bioenergy) underlining "Biomass Production for Energy from Sustainable Forestry" in Task 31 [11]. Finally, Agenda 21 calls for the improved management of forests to produce a variety of services, including "wood-based energy", and urges the greater development of renewable energy from forests and agriculture [12].

The crucial problem is then of how to combine biomass production with sustainability. What does sustainability mean in the wood production context? How can it be designed, implemented and monitored?

This article is organized around three core questions about integrating sustainability in the bioenergy sector: the "Why?", "What?", and "How?". I first explain why wood-based bioenergy is growing in relevance, and why it has to be sustainable to serve its purpose. After briefly highlighting the potential tradeoffs between sustainability and economic affordability of bioenergy, I ask what sustainability ought to be, and highlight the different ways in which it is understood. A critical review of the literature shows that while sustainability is a key aspect, it remains under-investigated, and often a hollow term. Most strategies for transitions towards bioeconomy are very broad frameworks, and do not go into the details of environmental and socio-economic sustainability [13]. It was observed that only one bioeconomy perspective, dubbed the 'bio-ecology vision' clearly focuses on sustainability, but that it is often dominated by the competing 'bioresource' and 'biotechnology' perspectives [14]. I categorize existing attempts of assessing sustainability in wood production for bioenergy into four levels, from "thin" to "holistic". I then move to the key question—how sustainability can be effectively mainstreamed. It is no coincidence that the forestry sector is probably the most experienced with the development of advanced sustainability frameworks. These need to be adapted to the growing role of wood in bioenergy production. Drawing on the sustainability governance literature, I show two complementary pathways for integrating sustainability in wood-based energy sector governance: top-down and bottom-up. The former emphasizes the role of international and globally applicable governance tools, such as agreements, certification schemes, and standards dedicated to woodfuels. Conversely, a bottom-up approach departs from local specificities and examples of already existing sustainable modes of production. To highlight how sustainability is achieved in the actual practice of forest plantations for energy needs, I employ project data gathered in Paraguay. These data, gathered through fieldwork, direct observation, and expert interviews, are used in a Multi-Criteria-Analysis (MCA) of a project's sustainability in three areas (social, environmental and economic). I conclude with some policy recommendations, which flow from the analysis, as well as directions for future research.

## 2. Why Should We Ask about Bioenergy Sustainability?

Biomass can have different sources of origin, but wood is the most common. It was the major source of energy until coal took over its role in the mid-18th century, and today, it still remains the dominant energy source for some 2 billion people [15]. Bioenergy from wood covers about 10 percent of the global energy supply, with about two-thirds being consumed in developing countries for cooking and heating purposes [16]. Global energy supply from forests was estimated to reach 772 Million tonnes of oil equivalent (Mtoe), 496 Mtoe from forests, and 276 Mtoe from forest processing [17]. Because biomass is difficult to quantify (estimates may be restricted to the aboveground vegetation, or only to trees or their components (such as foliage, wood, bark, and branch wood). Wood fuel

on the stand is often measured in tonnes removed from the area. There is a significant difference between solid cubic meters and loose chips cubic meters, while the amount of energy largely depends on moisture), and because fuelwood is mostly produced and consumed locally and often informally traded, this greatly affects the availability and accuracy of woodfuel statistics [8].

Wood biomass is particularly important for household cooking and heating purposes in the developing countries [12,18], but it is also a large part of energy mixes in the global North. In the United States, wooden biomass consumption almost equals biofuel consumption, and it constitutes some 20% of all renewable energy consumed [19]. In the EU, 46 percent of renewable energy originates from solid biomass that is almost exclusively wood alone. Biomass energy consumption increased from 72 Mtoe in 2004 to 128 Mtoe in 2013 [16]. Solid biomass and charcoal have provided 63.6 Mtoe of energy available for final consumption in 2013, and that market is constantly growing, with Germany, France, Italy, Finland, and Sweden in the lead [20].

However, while it is recognized that woodfuels have significant potential to address energy needs, offer fossil-fuel alternatives or bring direct benefits to rural areas, there are still many sustainability concerns (see Table 1). These trade-offs, which might not be clearly visible for policymakers beforehand, are the reason for why we need to mainstream sustainability thinking into bioenergy discussions.

**Table 1.** Areas of concern: General possible effects of woodfuel production (Source: own elaboration, based on [9,18,21–27]).

| | **Categories of Effects** | | |
|---|---|---|---|
| **Impact** | **Social** | **Economic** | **Environmental** |
| Positive | Rural employment<br>Infrastructure development<br>Improvement in community access to energy<br>Value-added products and credit facilities<br>Rural development<br>Poverty reduction<br>Acquisition and transfer of technology | Income generation<br>Economic leakage<br>Economic diversity and resilience<br>Accessibility and affordability of woodfuels<br>Diversification<br>Infrastructure development<br>Access to energy for families and small enterprises<br>By-products | Sustainable energy<br>Renewable Energy Sources (RES)<br>Clean energy<br>Climate change mitigation<br>Carbon sequestration<br>Improved soil quality in comparison to agricultural food crops<br>Wood ash may be applied as fertilizer |
| Negative | Working conditions<br>Migrant labor<br>Child/forced labor<br>Land ownership/access to land<br>Community and cultural dilution<br>Impact on social organization and demographics<br>Impact on health | Imbalance of economic benefit<br>Increased competition for biomass<br>Woodfuel price increase and reduced accessibility<br>Food and forest product prices increase | Exposure of soil surface, soil compaction<br>Reduction of soil organic matter, reducing nutrients<br>Negative impacts on ecosystem hydrology<br>Changes to physical water properties<br>Changes to chemical water properties<br>Changes to biological water properties<br>Land use change, decrease in forest cover and habitat connectivity<br>Carbon emissions from land use change<br>Loss of ecosystems (grassland, natural forests)<br>Decrease in habitats<br>Species loss<br>Decrease in genetic diversity<br>Carbon removed through harvesting<br>Air pollution<br>Waste<br>Transport with fossil fuels |

Increasing wood demand can put pressure on forests; lead to deforestation, and negatively affect wildlife, soils, water cycles, and climate. Wood is a finite resource, and forests have multiple functions to perform. Tapping forests for biomass increases the competition over wood resources, and

becomes similarly controversial as food crop production for biofuels [18]. Many non-governmental organizations (NGOs) oppose the use of whole trees coming from natural forests in energy production, some support planted forests for energy purposes or wood from thinnings and selective cuttings. In the waste hierarchy pyramid [26], treatment for energy is among the least preferred options. In an ideal world, wood should be burned at the end of a long value-chain, when there are no better resource uses.

Furthermore, while biomass emits less $CO_2$ than coal, it is still hardly a low-carbon solution. Especially co-firing (a.k.a., co-combustion) of wood biomass with coal, which the EU recognizes as renewable generation, raises many concerns regarding economic and environmental performance, emissions, and perpetuating carbon lock-in [27–29]. Co-firing currently helps numerous EU countries (like Poland, Latvia, and Finland) to achieve their 2020 Renewable Energy targets. Bioenergy is often assumed to be carbon neutral, because of the carbon dioxide uptake in plant growth. However, there are huge uncertainties and challenges in estimating carbon emissions, and indirect land use change can significantly increase greenhouse gas (GHG) emissions, not to mention land conversion effects on the biodiversity [30]. Land use change contributes about 15% to global emissions of greenhouse gas, and it is estimated that 3% of global agricultural land is currently dedicated to bioenergy production [30] (p. 562).

Significant risks are also identified in the social and economic realms, wood-fuel international trade expected to rise significantly may leave its producers with little benefits and all negative side effects [24]. In many developing countries, the flourishing traditional biomass industry lacks sustainable management, efficient processing, and end use. Large-scale biomass development, e.g., in Africa, raises concerns about local land-use conflict and food security [9].

With the growing world population and energy needs, dependence on fossil fuel supplies, decreasing rural employment, energy from wood needs to become part of the solution, not a problem. For that to take place, it has to be sustainable. But what does sustainability mean in the bioenergy sector? The common conceptual base for sustainability comprising three dimensions—environmental, economic and social—originated in the work of the Brundtland Commission [31]. This is the triple bottom line approach, where a minimum performance needs to be achieved in all of the three pillars. The Rio Declaration emphasized that sustainability must go beyond the environmental aspects, also assuring social and economic benefits [32]. And yet, these general ideals are debated in practice since then [33] and various proposals on integrating the three pillars exist, e.g., the so-called 'three circles' model, 'triple bottom line' model, or the 'bullseye' model [34].

Dauber and colleagues state that while the expansion of wood bioenergy is "advanced", it is "not necessarily sustainable" [35]. The sustainability of the biomass is a complex and multifaceted issue, as "bioenergy is probably the most complicated type of RE for measuring sustainability" [36]. In the short value chains in developing countries, even estimating volume levels is imperfect [37], much less establishing whether the production processes are sustainable. This makes sustainability assessments very problematic and difficult to compare. A critical reading of a wide corpus of literature on woodfuel production helps to see differences in the way in which researchers approach this topic.

## 3. What is Sustainable Bioenergy? Four Levels of Sustainability Assessment Frameworks

The Joint International Energy Agency and World Bank report points to several reasons for why measuring and tracking the sustainable use of bioenergy is extremely complex. Firstly, it is due to different dimensions of sustainability (economic, environmental, and social) and their own indicator sets. Some scholars point to the complexity of the term, at least 300 various definitions of sustainability exist only in the environmental management-related disciplines [38]. Similarly, journalists criticize the terminology as blurred and ambiguous, spanning different semantic fields [39]. The multiplicity of ways in which sustainability is defined and understood "hinders the achievement of sustainability transitions" [40]. Secondly, it is context-specific, and assessments are prepared for different units (zones, project, or regions) difficult to translate to political units (country-level). Thirdly, data necessary

for assessments are intensive and expensive. Lastly, assessments need to be frequent, and monitoring progress and requiring institutional structures and data collection platforms [36].

Sustainability is often implied rather than really assessed. While bioenergy from wood is treated as a sustainable and renewable energy source, many institutions, including the European Parliament, have noted that in order to avoid negative climate-related effects (e.g., releasing carbon stocks through deforestation) sustainability criteria, taking the lifecycle of greenhouse gas emissions into account, are needed [16], and Life Cycle Assessment (LCA) is commonly used to estimate the net greenhouse gas emissions in bioenergy production [41].

Though in articles conducting sustainability assessments, it is not always easy to distill how the authors define sustainability-related concepts, there are nonetheless some general patterns. Clearly, some of what goes under the notion of 'sustainability' in the literature is not fully sustainable, while very detailed accounts might be impractical. Attention can be drawn to the dichotomy between 'weak' and 'strong' sustainability [42], or what I call 'thin' and 'holistic' approaches. The categorization below proposes to think of different 'levels' in sustainability thinking in woodfuel production (Figure 1). It is a classification that starts with instrumental 'thin' sustainability use that expands sustainability frameworks for more comprehensive assessments.

### 3.1. Level 1: Thin Sustainability: Wood as a Renewable Resource is Always Sustainable

Firstly, and not very uncommonly in the literature, woodfuels are automatically treated as 'a sustainable form of energy', simply because they are a renewable resource [43]. Here, renewable energy appears as a synonym for sustainable energy [44,45]. Similar claims are also made, regarding other resources, as agricultural products, where the renewable character of the resources should imply their sustainability [46]. In this approach, sustainability is always implied and not measured, and there are no attempts to investigate how the wood was produced. This is a potentially dangerous oversimplification which should be better addressed by researchers, and it results mostly from the almost complete detachment of a techno-economic perspective on energy from socio-economic and environmental concerns, which are at the heart of sustainability thinking.

### 3.2. Level 2: Balanced Management Sustainability: Assuring Regrowth

One could argue that real sustainability thinking in woodfuel production starts with the assumption that wood regrowth must be assured. Scientific forestry started from a narrow idea of wood yield and still, "the economic dimension of sustainability concerns the ability to maintain the sufficient production capacity of forestry to meet current and future demands for forest products and services through using the resources efficiently" [47]. The increment factor is the key for controlling that no more wood can be cut than is annually re-growing. This rests on a key indicator: the level of sustainable cutting. For example, in Bailis et al. [48], in the assessment of pan-tropical woodfuel supply and demand, 'sustainability' is equated with one simple indicator: whether or not annual harvesting exceeds incremental re-growth. Nonetheless, they estimated that between 27–34 percent of woodfuel harvested was unsustainable [48] (p. 267).

The idea that sustainability means assuring regrowth is mainstream, both among foresters, bioenergy experts, and policymakers. In many contexts, especially in developed OECD countries, this may work just fine. Under a transparent national governance framework with robust environmental and labor regulation, and with clear land ownership rights, sustainability can effectively equal regrowth. However, take just one of the above elements out, and the sustainability of biomass production can be questionable. Also, there might be environmental functions that an industrial forestry paradigm cannot capture, e.g., regrowth can be assured, but at the cost of biodiversity [49].

Many factors (the ecosystem services, role of rotation lengths, canopy structure, habitats) imply that sustainable forest harvest should be below 100 percent, but more research is needed to determine such levels for different forest types [50]. Exemplary research from Nepal used 80 percent of the Mean Annual Increment (MAI) as a sustainable harvest level [51]. Although regrowth is a crucial factor,

it should be only seen as the beginning of a sustainability assessment, since that concept covers all the environmental, social, and economic dimensions. This might be closer to the green growth concept, but with an on emphasis economic and environmental pillars [52].

### 3.3. Level 3: Two-Pillar Sustainability: Broader Sustainability Frameworks

Confronted with the complexity of sustainability, authors look for different frameworks to operationalize the concept. While many studies move beyond the simple regrowth minimal criterion, in the case of woodfuel studies, the indicators are often quite randomly selected. Muizniece and Blumberga [53] use the notion of sustainability to expand beyond environmental/climate concerns, and they add the dimension of national economic sustainability. Sustainability can be seen as a combination of renewable energy, climate mitigation benefits, and value added for regional economy [54]. On the whole, where sustainability is treated seriously, it is often mostly its environmental aspect as "research is still dominated by green/environmental issues" while "social aspects and also the integration of the three dimensions of sustainability are still rare" [55].

A possible way to integrate more than one kind of factor is through formal tools such as LCA which was mostly applied to other woody products (furniture, paper, bioethanol), but only limitedly to the environmental profiles of solid woodfuel production [56]. Examples of LCA analyses in this sector build on different sets of impact categories. Fitzpatrick [57] proposes six categories: climate change, terrestrial acidification, freshwater eutrophication, photochemical oxidants, particulate matter formation, fossil depletion; while Laschi et al. [56] introduce seven: climate change, ozone depletion, terrestrial acidification, freshwater eutrophication, marine eutrophication, photochemical oxidants, and fossil depletion. In both cases, however, the impact categories are only environmental, though LCA does open the way for multi-criteria assessments as well [58].

Some studies go against that current, putting more emphasis on the social aspects of sustainability, e.g., combining a balance between supply and demand with assuring participation [59], or combining regrowth (halting deforestation) with public health issues of biomass (indoor air pollution because of woodfuel use for cooking), as well as concerns for soil quality, desertification, species loss [60], or food security [61], female labor force participation [62], or household incomes [51]. This current of research echoes more normative and political theoretic work in energy studies, which has recently seen the development of concepts and frameworks of energy justice [63], emphasizing fairness and social inclusion, or energy democracy [64,65] which focus on political participation and ownership. What is lost, however, is the environmental dimension—in all the above-discussed frameworks that were undermined by socio-economic concerns.

### 3.4. Level 4: Holistic Sustainability: Introducing Frameworks across all Three Pillars

The last stage, or the most extensive level of sustainability integration in analyses and assessments, are those that integrate all three 'pillars'—economic, social, and environmental. In terms of underlying ideas, some studies in forestry and bioenergy go even further, emphasizing the additional environmental benefits of 'cascading use', meaning a sequential use of biomass from the highest values, to energy recovery at the end of the life cycle [16]. This moves beyond the paradigm of bioeconomy, and into a new realm – circular economy, where the emphasis is not only on the biological origins of resources, but also on their fullest possible and preferably continuous use (the promotion of the use of wood waste for energy), coupled with low environmental impact, creating economic value and jobs. Definitely, it is imperative to extend wood life cycles where possible. However, it is no less important to strive for a sustainable production process.

Multiple Criteria Decision Making (MCDM) can be applied to address the multiple goals of sustainability. An extensive review study lists 30 examples on what technical, social, economic, and environmental criteria were applied into various MCDM models. Interestingly, there are examples with an assessment of only two indicators (economic criteria and reliability in the case of no. 15) contrasted with assessments taking into account 14 indicators [66]. Similarly, Szulecka [67] applies

a formalized Multi-Criteria Analysis (MCA) framework with 47 indicators across social, economic, environmental, and technical areas.

Büyüközkan and Karabulut [68] propose a novel sustainability framework for energy project performance evaluation, including biomass energy, based on MCDM and Analytic Hierarchy Process (AHP) perspectives, and listing 37 criteria for evaluation. Hansen et al., propose 12 criteria that are observed in Sweden and Norway and instituted in legislation and industry standards to ensure the sustainability of bioenergy value chains [69]. Another ambitious example is the sustainability index for global biomass, based on 27 parameters comprising social, economic, and environmental indicators developed by Barros et al. [7]. Sustainable Energy Security (SES) with four dimensions: availability, affordability, efficiency, and environmental acceptability (the last interpreted as a lower use of resources, reduced waste, and reduced emissions) offers an interesting blend of sustainability and energy security [70]. Broader analytical comparative frameworks for assessing sustainability initiatives have also been developed with thematic axes (top-down vs bottom-up, ecological vs socio-economic, holistic vs subject-specific, and regional vs local) to measure and balance the performance of given sustainability initiatives [71].

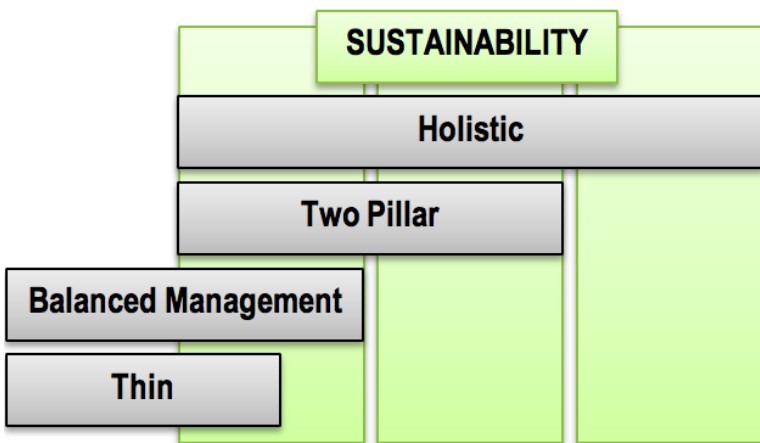

**Figure 1.** How much sustainability in 'sustainability'? Comparing four levels of assessment frameworks. The vertical 'pillars' represent three dimensions of sustainability (economic, social, and environmental), the horizontal blocks illustrate which assessment frameworks take these into account.

While it is important to be aware that 'sustainability' means different things in different studies and policy frameworks, unnecessary complexity is not the goal here. The four levels of sustainability integration in analytical frameworks are only meant to show the difficulty of moving sustainability from a vague political idea to an operational indicator, which is to achieve concrete environmental, social, and economic benefits. Simple analytical frameworks can also be useful, but they need to be more comparable. As the above review shows, no two studies use the same measure of sustainability, making the answer to the supposedly simple question of 'what is more sustainable?' across cases virtually impossible. Sustainability literature rooted in environmental studies and political ecology needs to respond to that, and enter technical debates within climate and energy studies. What is more, pressing is the practical question: how to mainstream more holistic sustainability frameworks in actual bioenergy governance, so that we can achieve more sustainable biomass production?

## 4. How to Make Bioenergy Sustainable? Two Governance Approaches to Sustainability Standards Integration

The governance literature suggests two possible ways regarding how a particular norm can be integrated into policy, and influence industrial and economic activity. One is a top-down approach that assumes that the mainstreaming of environmental norms in 'global environmental governance'

requires some sort of an agreement at the (inter)governmental or trans-national level (then also including non-governmental, possibly private actors) [72–75].

The alternative view has a bottom-up logic focusing on the local actions and realities, and the way in which these can serve as examples, be replicated and expanded in other contexts [76–79]. The top-down paradigm is stronger in defining the system boundaries, general context, and conditions, while it is the bottom-up that provides the real context and competing visions, goals, and scenarios. Some authors treat both levels separately, and some try to push for an adaptive learning process taking both perspectives into account [80–83].

The top-down approach usually focuses on global forestry negotiations and formal governance tools, such as treaty enforcement, certification, and standard-setting. On the other hand, the bottom-up perspective is micro-focused, with case studies pursuing more nested and often local forms of knowledge, focusing on local actors, specific forestry management systems and practices, assuming that such local anchoring is especially important for solving social and environmental problems at the heart of sustainable development. This dual approach is also typical in climate change mitigation and adaptation measures [84,85].

Both perspectives in the real-life context are largely complementary: top-down standard setting should be designed with adequate input from below, and local actions should promote adequate solutions at the global scale. Consequently, promoting sustainability requires a dual process of dialogue between macro- and micro- levels. Nevertheless, how much the two levels interplay, depends on many factors, including the capacity of institutions, their resources, and power to design and enforce sustainability norms.

The following empirical sections apply both these approaches. From the top-down perspective, I look at the emerging sustainability frameworks, and provide a mapping and comparison of existing standards to understand their strengths, limitations, and blind-spots. For the bottom-up perspective, from a wide set of cases that can serve as an illustration for sustainability enhancement in woodfuel production locally, I selected an exemplary plantation project organized as an out-grower scheme. It illustrates a situation where sustainability has to be enforced where there are no external (global or national) incentives.

*4.1. Top-down Approach: Assuring Sustainability*

4.1.1. Global Forestry Governance Tools

Contrary to biodiversity, climate change, or combating desertification, there is no global binding convention on forestry. As governments are not able to agree on universal and binding forestry norms and rules, it is the NGOs, civil society, and the private sector that have pushed for the international sustainability standard development that is supposed to fill this gap [86]. As a result, some of the "most extensive and innovative experiments in 'new' governance" [87], are in the forestry sector, with certification as a primary example.

Third-party certification initially showed a great potential to supplement the ineffective global forest regime [88]. The two most significant certification schemes are the Forest Stewardship Council (FSC), established in 1993 [89], and the Programme for the Endorsement of Forest Certification (PEFC) established in 1999 [90]. A powerful tool in bringing many sustainability issues into careful consideration, certification also has limitations, most importantly—its voluntary character. Others are the costs of certification, methodological, and auditing problems. It was also observed that producers as a group are often not equally represented in the Voluntary Sustainability Standards-Setting Initiatives [91]. Certification has also allowed for free-riding, leading to a race to the bottom in standards. Industry-driven competing schemes, less rigorous than FSC, have emerged in many regions, and have started to take over the FSC targets, often watering down their normative prescriptions [86,92].

Nevertheless, certification and verification schemes can contribute to the sustainable production of bioenergy. Though it is often considered a technical issue in a narrow sense, certification should be understood as a governance tool and a framework for debates [21]. Current schemes can be an important anchoring for assuring bioenergy sustainability, as they already cover many woodfuel-related aspects. For example, pellets and briquettes are woodfuels that can be tracked in the FSC and PEFC product databases [25]. However, there still remain significant gaps and problems, e.g., related to the definitions or the evaluation of specific impacts in woodfuel production systems.

### 4.1.2. Designing Criteria and Indicators for Wood Biomass

There is no direct multilateral agreement that is dedicated to fuelwood [12], existing policies are divided, often informal, and contain significant gaps [93]. Even in the EU, there are no binding sustainability criteria for biomass at the EU level, e.g., the Renewable Energy Directive 2009/28/EC does not offer any [94], though some exist at industry and national levels [16]. EU Forest Law Enforcement, Governance and Trade policy differentiates between legal and sustainable resources [95]. Therefore, the sustainability of biomass imported to Europe seems a blurry objective. The emphasis is on the legality of wood (targeting illegal logging), hard to enforce but possibly easier to regulate than social, economic, or environmental sustainability standards.

A possible solution for implementing woodfuel standards is in streamlining of bioenergy criteria with existing sustainable forestry indicators and forest certification [25]. Sustainability standards are among the most advanced in the forestry sector [96,97], which "has been at the vanguard in introducing the concept of sustainability" through Sustainable Forest Management (SFM) [56].

There exists many other schemes relevant for biomass like Criteria and Indicators (C&Is) for SFM by Forest Europe, certification schemes discussed in the previous section, industry-led initiatives (sustainable biomass partnership, ENplus certification for wood pellets) [16]. There are also specific biomass, bioenergy, and woodfuel-dedicated frameworks, as the sustainability principles of the Round Table on Sustainable Biomaterials (RSB), FAO standards, or sustainable bioenergy indicators developed under the Global Bioenergy Partnership (GBEP).

Table 2 shows regulatory and standard setting initiatives that aim to assure sustainability in wooden biomass production. They have various geographical foci, come from different fields (biofuels, woodfuels, bioenergy, forestry, etc.) and have different degrees of comprehensiveness. It is also visible that different levels of sustainability are possible, and it needs to be considered separately in the social, economic and environmental realms, as discussed in the Section: 'What is sustainable bioenergy'.

**Table 2.** Exemplary regulatory and standard-setting schemes and initiatives compared (Source: own elaboration, based on [9,12,81–86,98–103]).

| | Roundtable for Sustainable Biomaterials (RSB) | Food and Agriculture Organization of the United Nations (FAO) | Global Bioenergy Partnership (GBEP) | Forest Stewardship Council (FSC) | Programme for Endorsement of Forest Certification (PEFC) | Forest Europe | Sustainable Production of Biomass—The Netherlands |
|---|---|---|---|---|---|---|---|
| **General Information about Existing Schemes and Initiatives** | | | | | | | |
| Scheme (name or acronym) | Roundtable for Sustainable Biomaterials (RSB) | Food and Agriculture Organization of the United Nations (FAO) | Global Bioenergy Partnership (GBEP) | Forest Stewardship Council (FSC) | Programme for Endorsement of Forest Certification (PEFC) | Forest Europe | Sustainable Production of Biomass—The Netherlands |
| Geographic scope of the scheme | Global | Global | Global | Global | Global | European | 20 countries |
| Operational since (year) | 2011 | 2010 | 2011 | FSC-STD-01-001 V5-0 2012 | PEFC ST 1003 2010 | 2002 | 2011 |
| Main focus of the scheme | Biofuel | Woodfuels | Bioenergy | Forests SFM | Forests SFM | Forests Sustainable forest management (SFM) | Biomass |
| **Degrees of Generality or 'Granularity' in How a Scheme Operationalizes its Standards: from most General (Theme) to most Specific (Indicator)** | | | | | | | |
| Themes | n/a | n/a | 3 | n/a | n/a | n/a | n/a |
| Principles | 12 | 4 | n/a | 10 | n/a | n/a | n/a |
| Criteria | n/a | 17 | n/a | 70 | 7 | 6 | 6 |
| Indicators | 37 | 57 | 24 | n/a | 66 | 35 quantitative 17 qualitative | 20 |
| Are there several sets of standards under the scheme? | RSB separate principles and criteria for smallholder groups <75ha | FAO separate principles, criteria, and indicators for sustainable charcoal production | No | No | No | No | An older version from 2007 |
| **Author's own Evaluation of the Degree of Sustainability Integration in the Schemes** | | | | | | | |
| Coverage of different sustainability aspects | Very comprehensive | Very comprehensive | Significant gaps | Very comprehensive but not woodfuel focused | Very comprehensive but not woodfuel focused | Comprehensive | Comprehensive |
| Additional comments related to sustainability issues | Focused on bioenergy, prescriptive $CO_2$ reduction of 50% compared to fossil reference, little emphasis on economic issues (does not mention, e.g., value-added or productivity) | Focused on woodfuels, does not cover several issues, e.g., training, infrastructure, air pollution | Does not cover many issues, e.g., legality, laws, monitoring, human and labor rights, rural development | Very well covers SFM, issues and many Criteria and Indicators (C&Is) apply to biofuels, but the scheme does not explicitly mention "energy" nor "fuel" at all | Very well covers SFM issues and many C&IS apply to biofuels, but the scheme does not explicitly mention "energy" nor "fuel" at all | Focuses on SFM, criteria, and indicators are descriptive, not prescriptive; 6.9 focuses on Energy from wood resources | Prescriptive $CO_2$ reduction of 50% compared to fossil reference, does not cover several issues, e.g., legality, monitoring, GMO |

Such frameworks operate at different stages in the bioenergy supply chain, and SFM schemes can be most readily transplanted into schemes for sustainable biomass, as they already focus on, e.g., the maintenance of production, soil fertility, habitat, species diversity, impact on soils, workers' rights and safety, or management and monitoring. Therefore SFM 'has the potential to play a leading role in defining sustainable woodfuel production in both the North and the South in both plantations and natural forests' [25].

Here we reach a very important caveat. It is estimated that strict sustainability criteria reduce the production potential of biomass by up to 30 percent [16]. This can explain why development for sustainability frameworks, sets of criteria and indicators is a slow process. Significant production loss prognosis and applied methodological challenges are hindrances in the process of sustainability standards development in the field. Biomass certification is also difficult to classify, falling under biofuels, forestry, or agriculture sectors. It can be also developed for specific purposes—fair trade or particular environmental concerns. Additionally, there are general methodological challenges in categorizing wooden biomass, C&Is use different terms, e.g., forest fuels, fuelwood, firewood, wood energy, biomass energy, forest products for energy [25]. FAO differentiates between woodfuels, agro-fuels, urban waste-based fuels and traditional bioenergy (firewood, charcoal, residues) and modern (industrial wood residues, energy plantations, use of bagasse) [1].

Another issue is the methodological challenge of assessing and monitoring sustainability progress: assessments need to rely on different types of indicators, qualitative, semi-quantitative, and quantitative, proxy, country-level, etc. Some indicators are easier to obtain (jobs in the bioenergy sector, allocation, and tenure of land for bioenergy production), while others may be extremely difficult to calculate (change in unpaid time spent by women and children collecting biomass or emissions of air pollutants or change in the diversity of total primary energy supply due to bioenergy). Also, some methodologies receive more scholarly and political attention, i.e., modelling greenhouse gas emissions developed more quickly as a proxy for energy sustainability, while most methodologies remain weak, for instance, in assessing the impacts of bioenergy on food security [36]. Still, this is not to deny the importance of modelling greenhouse gas emissions, or challenges and complexities that are linked to estimating emissions from land use change. Particularly, including indirect land use change in lifecycle emission accounting poses enormous challenges in identifying, interpreting knowledge, and using it in policy-making [104,105].

Are certificates and their C&I frameworks a one-size-fits-all solution? Not necessarily. Developed countries put more emphasis on standards related to the environmental consequences of harvesting on soil and biodiversity, while developing countries underline social and economic aspects of sustainability [25]. Additionally, some scholars point out that the environmental sustainability of biomass production systems often focuses more on the global effects for the environment (greenhouse gases and pollutions) not local environmental impacts, e.g., on soils or biodiversity [88]. Certificate scheme implementation varies between regions. PEFC and FSC show significant difference in reaching the developing and developed countries, and when there is no value-added to the wood products and especially no exports, there exists little incentive to adopt stronger sustainability standards. There are also practical problems with standard enforcement, monitoring, and auditing. Finally, we reach the ethical question regarding development capabilities, and how certification should be designed to avoid disadvantages to smallholders by, e.g., unjustifiable administrative or financial burdens [22]. Even sophisticated analytical tools and standards are always based on normative judgments. Sustainability should be perceived both globally and locally. Therefore, the bottom-up approach is equally important.

*4.2. Bottom-up Approach: Promoting Sustainability*

As noted already, when woodfuels are produced in the global South and they have a short value chain without exportation and are not targeted by developmental projects, adoption of specific woodfuel standards is highly unlikely, unless it is required by the national legislation. Therefore, it is as important to look for interesting bottom-up initiatives towards more sustainable bioenergy to act as

'role models'—practically tested and locally anchored solutions which can be scaled-up or promoted in other contexts [42].

### 4.2.1. Woodfuels in Plantation Systems

Unlike industrial timber, woodfuels come from different production systems, including natural and planted forests, wooded village areas, trees outside forests or in combined systems in agroforestry or silvopasture. Important production systems for woodfuel energy are short- and medium-rotation coppice (SRC and MRC) [106]. Trees (typically willow, poplar, and eucalyptus) are grown closely, and harvested in short periods [107,108]. The technique was developed in Northern Europe, but it has received significant attention in other countries, including Germany and Italy [109,110]. Categorized by the ownership, there are two popular plantation types: large-scale industrial and smallholder. Both sharply increased in volume in the recent decades [12,111], and have a potential to bring the 'plantation conservation benefit' [112], and both have their own "typical" sustainability problems. Industrial plantations in the global South may induce social conflict [113–115], bring expropriation, cause adverse environmental effects [116], biodiversity loss [117], and result in massive economic failure [118], all leading to questioning their potential sustainability unless strict regulation is applied. At the same time, small-scale social forestry projects and community-based approaches [119] alone cannot play the role that is envisaged for global wood supply, and face their own problems for decades: market access, safety standards, pest management, access to technology, etc. The two might seem irreconcilable; however, for sustainable woodfuel production, finding a mid-way between the two approaches is not only possible, but a necessity. One option is in corporate–societal partnerships like out-grower schemes.

### 4.2.2. Woodfuel and the Corporate/Societal Nexus: Out-grower Plantation Schemes

Bioenergy projects can lead to different positive or negative environmental and socio-economic impacts, creating both opportunities and risks (Table 1). Risks are more significant in the developing countries, often lacking adequate legal frameworks and enforcement. Nevertheless, due to the global nature of the bioenergy market, the worldwide scope of energy companies, and better growth rates in (sub)tropical regions, sustainability analysis must focus especially on those areas.

In order to minimize risks, bioenergy producers should be aware of and rewarded for their good practices. If implemented well, out-grower programs are among the potentially best practices that reduce the risk of unsustainable use and that harness opportunities in bioenergy [36,67].

Out-grower schemes are examples of a corporate–societal "contractual partnership between growers or landholders and a company for the production of forest products" [120], meaning arrangements between private-business or cooperative-industrial actors (capital), and local communities (labor and land). Such partnerships have been proposed in different areas of land use for several decades, sometimes also called 'partnerships' or 'contracted farming' [120–128]. Though mostly discussed in developmental contexts, such partnerships exist elsewhere. In the Nordic countries, public–private partnerships involving different stakeholders such as biomass producers, industries (e.g., sawmills, pulp, and paper plants), entrepreneurs, public authorities, and NGOs also exist, and are becoming increasingly popular and more extensively studied in the sustainability context, as a model of a bioenergy community [129]. A similar trend of emerging public–private partnerships and the development of agro-energy district models is documented in Southern Europe, as studies from Greece and Italy show [130,131].

Research on out-grower bioenergy plantations is scarce and fragmented. Hoffman et al. investigated out-grower schemes for energy production in Tanzania, and focused on sustainability indicators for small-scale farming of bioenergy crops [132]. FAO gives examples of good practices in bioeconomy, decentralized implementation mechanisms, by analyzing the Malaysian Community Development Programme that promotes out-grower schemes [13].

If designed and implemented well, out-grower schemes might be a solution to harmful land-grabbing practices, lead to good income generation [124,133], increase productivity [134], act as the local saving banks [135], facilitate technology transfer and market access to the growers [125], lead to inclusion, community development, improvement in local access to energy and energy security. They can further increase environmental awareness and good environmental practices, adding to the development of social capital and technology transfer that is anchored in the local institutions [136], climate change adaptation and mitigation, contribute to better soil quality, water availability, and agrobiodiversity [137,138]. As for the investors, it is an attractive solution to secure land and a workforce for an investment that will not bring inherent conflicts [139]. However, for the arrangements to be effective, they need to be carefully designed and implemented to avoid adverse impacts [140].

### 4.2.3. An Empirical Illustration: a Bioenergy Out-Grower Scheme in Paraguay

To draw on an existing example of successful sustainable wood-based energy production practices, which illustrates the 'bottom-up approach', I point to a project developed by the farmers' cooperative, Colonias Unidas, in Paraguay. Although out-grower schemes are partially known to development practitioners, foresters and land use scholars, there are still neglected in the climate policy debates. Research conducted on-site shows that an out-grower scheme can generate a range of sustainability benefits for stakeholders. The Multi-Criteria Analysis (MCA) employed here illustrates how a holistic model of sustainability assessment can be operationalized in woodfuel research, and how important it is to develop comparative frameworks that visualize the tradeoffs and that help to learn from comparative assessments.

The cooperative Colonias Unidas started its internal plantations out-grower scheme contracts for bioenergy purposes in 2000. The out-grower program was aimed at covering the cooperative's energy needs: 25,000 tons of firewood annually. A special credit line was designed for smallholders to make woodfuel an attractive production choice, especially favored for remote and marginal lands [141–145]. The project is a truly local initiative, developed in a 'blind spot' area, where international certification and verification schemes do not reach.

The scheme's sustainability was assessed in a Multi-Criteria Analysis, using a holistic approach to sustainability, building on a general appraisal combining field interviews, a study visit to the cooperative, and an analysis of the cooperative's documents. The results of MCA provide a stand-alone assessment of the project's sustainability (see Figure 2). The reason for why this project is used as an illustration and offered as an example of effectiveness is that a comparative analysis (using the same MCA methodology based on 16 clusters of indicators) was conducted for several other smallholder plantation projects in Paraguay [67], and the out-grower scheme in Colonias Unidas has scored highest on all indicators. In terms of social (90 percent) and economic (84 percent) dimensions of sustainability, it clearly outpaces other production systems oscillating between 50 and 60 percent levels [67]. It scores very well on participation, access to services, human resource management, and workplace safety. Its economic performance is also successful, both in absolute terms (good market access and profitability), but also in the redistribution of economic benefits, and in providing a good environment for investments, meaning stability and security. However, Figure 2 also clearly shows that performance in social and economic dimensions was much better than in environmental (56 percent). However, when compared to the analysis with an industrial *Eucalyptus* growing company the environmental score with the same indicators was significantly higher to the score obtained by the company or smallholder plantations [67]. The plantations often replace intensively managed agricultural land, and they are perceived as positive for the land use change and water regime.

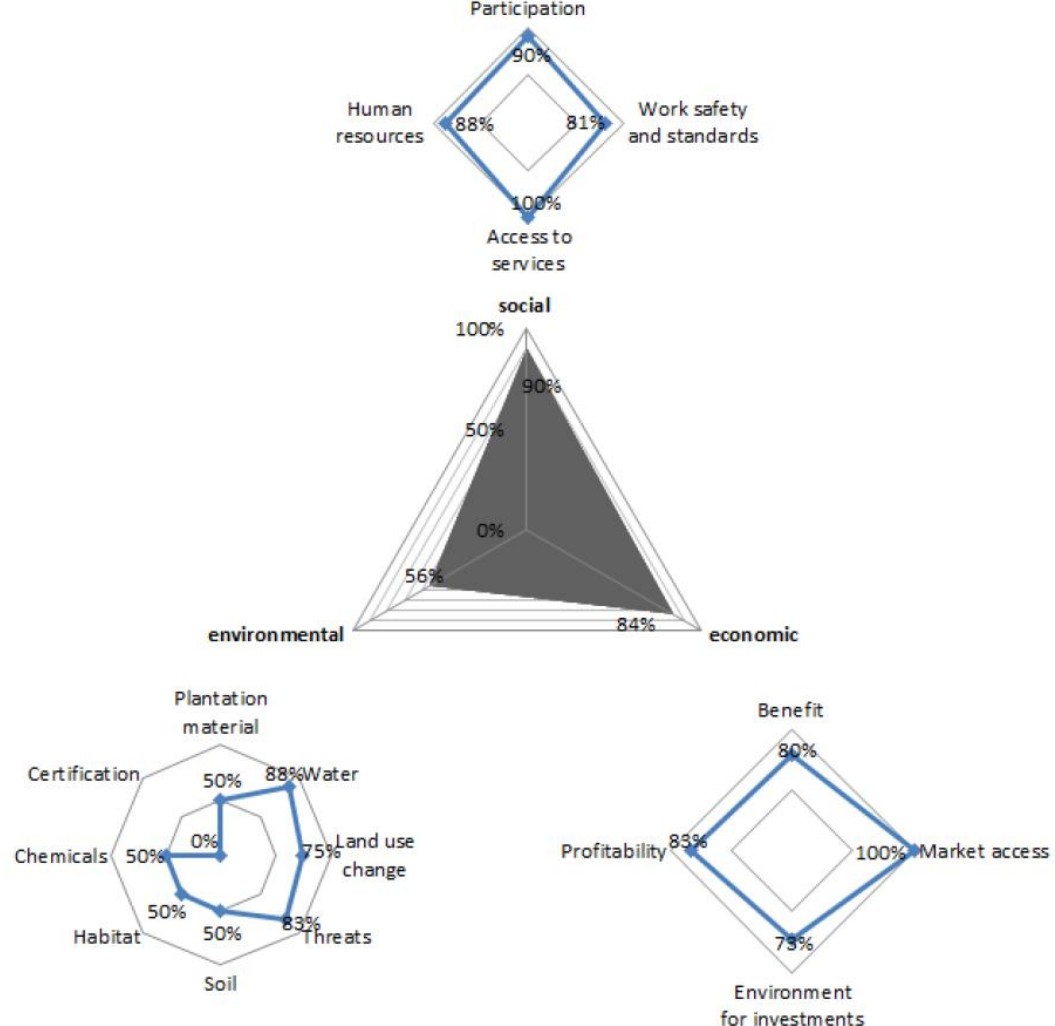

**Figure 2.** Multi-Criteria Analysis of out-grower schemes in Colonias Unidas (Source: [67]).

One weakness in terms of environmental performance, according to the way in which the MCA was operationalized, was the lack of certification for the wood produced. We have to note that the lack of a top-down sustainability tool (certification scheme) was a criterion for case selection, and should not be used against the studied plantation (the criterion was nevertheless kept to compare the case with other production systems in Paraguay).

The out-grower scheme case shows that such partnerships have a potential to increase sustainability in different aspects. Furthermore, using aggregated and visualized sustainability indicators illustrates what sustainability really means, and it helps to easily diagnose where it should be improved.

## 5. Discussion and Conclusions

As bioenergy from wood gains political currency, the problem of its sustainability becomes more relevant too. Ensuring that current and future woodfuels are developed in a sustainable way and benefit all affected stakeholders is challenging. The presented analysis of both sustainable bioenergy scholarship and governance leads to two types of conclusions.

There is no 'one-size fits all' solution, and as the analysis shows, both top-down frameworks and bottom-up sustainability initiatives must complement each other. Many European countries express, on the one hand, the need to develop supranational top-down sustainability tracking and accountability, rather than a national one. This is due to the global dimension of bioeconomy. On the

other hand, we can witness the proliferation of bottom-up initiatives that are led by the civil society, industry or regional cluster developments [13].

What is crucial is that sustainability ought to be taken seriously, and that implies both criteria/indicators and assessment frameworks (also useful for project evaluation) which are on the third (Two Pillar) or fourth (Holistic) level in the presented analysis (Figure 1). There are good reasons to argue in favor of "slowing down with political pressure and ambitious quotas and targets until full sustainability appraisals are established" [35]. Without these, policy discussions of sustainability remain cheap talk. The process of developing indicators should also be seen as production of knowledge, fact and value-based considerations [52].

In practice, the environmental pillar of developed projects often needs strengthening, as shown in the out-grower scheme analysis. Also, if we take the nested ring model of sustainability [34] the environmental pillar is a fundament for the social and economic pillars.

A well-designed and implemented out-grower scheme might be an attractive production system for wood energy production and sustainability standard implementation. Nevertheless, there always should be a third party involved in the partnership agreement as a guarantee for mutual duties (state agencies, NGOs) to limit the risk that the out-grower scheme will be used as an alternative form of land rent and companies' access to new financial schemes. This again highlights the importance of comparable and well-designed assessment frameworks.

More systematic research and comparative studies are needed on woodfuel sustainability—both technical and statistical, as well as normative and ethical. A critical review of sustainability assessment frameworks shows that more dialogue is required between sustainability scholars and technical disciplines to create a common ground for evaluations. While sustainability scholars have a lot to teach their technical colleagues about designing explicit ultimate goals, they have much to learn about possible assessment tools. The ideal for sustainability assessment is to have a holistic view, building bridges between the methodologies like the recent attempts to broaden the environmental pillar focused Life Cycle Assessments with social factors (S-LCA) [146].

Policymakers look to scholars not only for assessments but also diagnoses and preference constructions [147] and for that, we need better data. We have many single-case studies, and we need a comparative analysis drawing on multiple case studies, broader reviews, and advanced models. Improving sustainability in bioenergy development also requires building knowledge and databases regarding environmental and socio-economic impactions of land use for energy production, also lessons learned from other biofuels programs [148]. This global data gathering can be combined with new pilot project 'experiments' on the ground [35]. One important area that requires further research is the scope and impact of existing and developed out-grower schemes in bioenergy, from a sustainability perspective. Finally, empirical research needs to contribute to theory building, to achieve a stronger theoretical basis [55].

**Funding:** This research was funded by the Research Council of Norway and its BIONÆR programme within the SusValueWaste project, grant number 244249.

**Acknowledgments:** The finalization of this paper was conducted within the framework of the project "Sustainable path creation for innovative value chains for organic waste products" (SusValueWaste). I would also like to thank the discussant and participants of the SusValueWaste conference in Frederikstad in August 2017 for their constructive comments, as well as the five anonymous reviewers for their valuable feedback.

**Conflicts of Interest:** The author declares no conflict of interest.

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
