# Peer review of "Towards Sustainable Wood-Based Energy: Evaluation and Strategies for Mainstreaming Sustainability in the Sector"

_sustainability, doi:10.3390/su11020493_

Round 1

Reviewer 1 Report

The resulting scientific novelty and originality of your topic is not high. The aim of the paper as well as its scientific contribution is unclear. Contribution to the field and depth of research are poor.  As an interested reader, I was left disappointed with how thin the description and analysis (An empirical illustration: a bioenergy out-grower scheme in Paraguay). The author ignores the large and very well developed literature of the problem “biomass co-combustion”. Conclusion with references? The paper requires extensive editing for grammar (e.g. in references „Journal of Ceaner Production”; “America.. Praeger: New”;  “An African Case Study. . 2009.” ; Bioenergy. ; 2011.; Environmental science and pollution research international). In my opinion the text is not suitable for publication in high-profile journal. The criteria for publication of scientific papers are of outstanding scientific importance, reach a conclusion of interest to an interdisciplinary readership etc. As is my practice, I have read it carefully and my conclusion is that it is probably not best suited to the Sustainability.

Reviewer 2 Report

Summary

This paper provides a broad summary of sustainability issues in wood-based bioenergy.  It begins with a general overview of global trends in woodfuel energy. Next, it provides a framework of four approaches to conceptualizing sustainability for wood-based bioenergy. Lastly, it closes with a discussion of top-down versus bottom-up approaches to ensuring sustainability for these fuels, including a short example from the author’s experience in Paraguay. 

The broad coverage of issues means that it does not go into great depth or detail on any one topic.  It is really more of an overview article than a research article. Nevertheless, I think that this paper could be helpful for practitioners or policymakers who are new to this field and need an accessible introduction to sustainability issues.  My comments below point to numerous small corrections and clarifications. 

General grammatical comments

Superscript numbers: In English writing, it is customary to place the superscript numbers for footnotes and endnotes after the final punctuation of a sentence, with no spaces. 

For example, this is how the paper punctuates it: 

Faced with the complex challenges of combating climate change and achieving development globally, policymakers increasingly see bioenergy as an important element of future energy mixes 1

This is how I think it should be edited throughout for English publication:

Faced with the complex challenges of combating climate change and achieving development globally, policymakers increasingly see bioenergy as an important element of future energy mixes.1

Endnote references: appear to have duplicate numbers 

Specific comments

Title: The title refers to “sustainable bioenergy,” but the analysis in the paper is specifically focused on wood-based energy, which is a smaller subset of bioenergy. I think the title, therefore, overclaims its coverage. I think it should be revised to reflects its narrower focus.

Line 26: I am not sure that everyone will agree with this basic premise that policymakers “increasingly” see bioenergy as an important element of future energy mixes.  For example, in the US, many renewable energy advocates have fought the expansion of both biofuels (in the transportation sector) and wood-based bioenergy (in the electricity sector).  There have also been many critics of bioenergy in the EU, as I know from many biofuels debates, and also in the developing world, as seen in criticisms of wood-based home energy in favor of either grid- or off-grid-based electrification. So, I personally disagree with this statement – unless the author is specifically talking about how certain segments of analysts can’t figure out how to make the math work on decarbonization without bioenergy plus carbon capture. This brings me back to this starting premise. Instead of starting with such a broad statement, I think the author should be more specific about the nature of pro-bioenergy debates. Alternatively, if the author wants to start out from that broad premise, then she should provide references that are more recent than 2007. 

Line 30: Author should add the word “also” to this sentence, as in “bioenergy, from solid biomass, can also bring positive impacts” (italics added here just for emphasis). 

Line 45: The dash should be replaced with commas, and the words in the acronym should be capitalized. Thus, the sentence should read “… flagship initiative, Sustainable Energy for All (SE4All), also recognizes….” 

Line 86: The author states that wood biomass provides almost half of renewable energy in the US, citing an NRDC website. This is incorrect. Wood biomass provides about one-fourth of renewables in the electricity sector, or one-fifth of renewables in the energy sector more broadly. See EIA statistics: https://www.eia.gov/totalenergy/data/browser/?tbl=T10.01#/?f=A&start=1949&end=2017&charted=6-7-8-9-11To be more accurate, I recommend that the author rewrite the sentence using EIA data, not the NRDC website. Further, the author should make sure they are clear about whether they are talking about all bioenergy (including biofuels and wood biomass) and also whether they are talking about the entire energy sector or just the power sector. 

Line 98: Check the punctuation around the superscript numbers, as it seems off. 

Line 110: Carbon emissions is a major concern about bioenergy. It would be useful to cite some statistics here to give a sense of the magnitudes of emissions that have been estimated. Moreover, I think it would be helpful to add a sentence that acknowledges the uncertainties and challenges in estimating emissions.  This has been the focus of several policy discussions in the US and EU, especially related to land use change emissions. You’ll see that this is a theme running through several of my comments. 

Table 1: This is a helpful table that summarizes various sustainability trade-offs. I think it would be more readable if it were more compact, with no (or at least smaller) spaces between the lines and also without the extraneous lines at the bottom of each column (although I will leave it to the editors to deal with that formatting in the end).  The only substantive comment that I think must be made is that the table doesn’t acknowledge the sustainability issue that’s been a HUGE controversy in the US and EU for biofuels: carbon emissions from land-use change.  The table does mention land use change but frames it as an issue of forest cover and land connectivity rather than greenhouse gas emissions.  I think that this issue should be acknowledged somewhere in this table, and I’ve also added further discussion below.

Line 126 (and throughout): If these are direct quotes, they should be in double quotation marks rather than single quotations marks. This applies throughout the rest of the paper as well. 

Sectioning beginning at Line 133:  This is a clear and useful hierarchy of sustainability concepts. I would just add that I think they could be numbered, e.g., Level 1: Thin Sustainability (focused on renewable resources) 

Level 2: Balanced Management Sustainability (focused on managed regrowth)

Level 3: Two-Pillar Sustainability (focused on broader

Level 4: Holistic Sustainability 

Line 244: “o” should be “a”

Line 246: In “Listing criteria for evaluation. Hansen …” should replace the period with a comma. 

Line 248: Instead of “No less ambitious,” I think the author means “Even more ambitious.” But, to be honest, I found this and the following sentence to be awkwardly constructed. 

Figure 1: I don’t understand this figure. I thought that all of these four levels were supposed to represent some degree of sustainability. If that’s the case, then why are the green vertical bars only crossing some of these concepts? Also, if this is a rank ordering, why do these boxes differ in two dimensions? What’s represented in the Y and X axes?  

Section beginning Line 273: As a policy person, I will say that dividing policy into top-down versus bottom-up is a very basic way of categorizing policies. Policies could alternatively be categorized by level (e.g., global, regional, national, local) or by policy instrument (e.g., regulatory, fiscal, etc.) or by type of policy instrument (e.g., price instruments versus quantity instruments). This doesn’t need to be acknowledged in the paper. But since this is a pretty elementary way of describing policies, I thought I’d just flag this for the author as something to be aware of.  

Line 284: The phrase “political economic trends” is out of place here. Political economic dynamics affect all levels of governance and decision-making from international down to local. The phrase should be deleted.

[After this, I ran out of time for the detailed line-by-line corrections. My apologies. But there are numerous other places where the author needed to spell-check or have a close reading for grammar and word choice.]

Line 378:  Again, I am going to bring up the topic of carbon accounting. The paper claims here that “some methodologies receive more scholarly and political attention, i.e. measuring greenhouse gas emissions developed quicker as proxy for energy sustainability while most methodologies remain weak e.g. in assessing the impacts of bioenergy on food security.” I’ll note that the sentence is an example of how the paper needs to be proofread carefully for grammar, as it’s missing three commas (after “i.e.” and before and after “e.g.”) and an article (should read “a proxy” instead of “proxy”). But leaving the fine grammatical points aside, the main point I want to make here is that this sentence glosses over the huge controversy around greenhouse gas emissions. If someone doesn’t know about this topic, they might read this and think it’s a simple and settled issue, and there aren’t further references added to help the reader delve into the complexity. But the fact is that greenhouse gas emissions can’t be “measured” – they can only be modeled – and there are significant intellectual and political battles around the methodologies that are appropriate for environmental policy. I think that the paper should at least add a couple of sentences that discuss how implementing environmental regulations based on lifecycle greenhouse gas emissions, while seemingly straightforward in theory, has run into challenges with estimating emissions from land use change. This is primarily an issue in liquid biofuel regulations but could become an issue in bioenergy more broadly. Here are two references about the complexities of this type of sustainability policy that I think would be valuable to add:

Breetz, H. L. (2017). Regulating carbon emissions from indirect land use change (ILUC): US and California case studies. Environmental Science & Policy77, 25-31.

Palmer, J. (2012). Risk governance in an age of wicked problems: lessons from the European approach to indirect land-use change. Journal of Risk Research15(5), 495-513.

Empirical illustration from Paraguay:  It would be helpful to have more explanation about what this case study adds to the paper. Line 448 simply says that the purpose is “To illustrate the potential sustainability benefits of a well-designed partnership.” Sure, illustrations can be nice. But this does not tell the reader why a deep-dive illustration is needed here but not in the other section on top-down approaches. Why is this illustration necessary? What do we learn from this illustration? What do we not learn from this single illustrative example?

Reviewer 3 Report

I really enjoyed reading this article. It is very well written and is extremelly ineresting.

The only suggestion that I can make is about Table 2. I had some difficulties to understand it. It would be a plus if the author could rearrange the table in order to facilitate reading.

Reviewer 4 Report

Dear Authors,

I have just reviewed your paper titled "Towards sustainable bioenergy: Integrating 2 governance frameworks". The aims of the paper are germane with Sustainability, in this form of article fits marginally with the international scientific standards, it could be more suitable as review. The paper is written with an appreciable English level. The contribution of this paper to the scientific knowledge is good. In the text there are some flaws, mainly related to the formatting and I suggest the corrections in the comments for the authors and also in the file attached.

1.    Please, I suggest also some citation referred to energy system from SRC or MRC plantations for a better presentation of the paper.

2.    I suggest to correct the problems highlighted about the references.

Reviewer 5 Report

General comments

The paper deals with sustainbility frameworks for the use of woody biomass. The topic is of interest and the methodologyu is correct. Results are interesting.

Specific cooments

- I would change the title into "Towards sustinable bioenergy from woody biomass using Multi-Criteria-Analysis"

- check the format of the paragraphs (and also sub-paragraphs), they should benumbered according to the author guidelines;

- page 1 line 34, you write "But it also signals some potential problems ...". I think here you should also consider studies on biomass sustainability like:

Buratti, C., Barbanera, M., Fantozzi, F., A comparison of the European renewable energy directive default emission values with actual values from operating biodiesel facilities for sunflower, rape and soya oil seeds in Italy (2012) Biomass and Bioenergy, 47, pp. 26-36

- page 1 line 37, you write "... biomass scores the lowest among ...", but you haveto clear which kind of biomass? Is it coming from forests? This would make sense. So please specify;

- page 2 line 52, you write: "...energy from forests and agriculture". I think about agriculture you have to take into consideration the agroenergy district model developed in:

Fantozzi, F., Bartocci, P., D'Alessandro, B., Arampatzis, S., Manos, B., Public-private partnerships value in bioenergy projects: Economic feasibility analysis based on two case studies (2014) Biomass and Bioenergy, 66, pp. 387-397 Manos, B., Bartocci, P., Partalidou, M., Fantozzi, F., Arampatzis, S., Review of public-private partnerships in agro-energy districts in Southern Europe: The cases of Greece and Italy (2014) Renewable and Sustainable Energy Reviews, 39, pp. 667-678 - table 1 check please the brackets - table 2 there are sone cells which are empty. In that case please write on them n.a. (not available) - the paragraph "An empirical illustration: a bioenergy out-grower scheme in Paraguay" is too short increase the detail of the analysis and the data provided (tables and figures). Because it is interesting. - please check the reference format (you have doublenumbers of reference. Should bean error.

Round 2

Reviewer 1 Report

The author has carefully reviewed all suggestions, and changes have been made and marked to the last manuscript where appropriate. Explanation and corresponding revisions prove sufficient to the mentioned concerns and render the manuscript acceptable for publication.

Reviewer 5 Report

The required changes have been performed. Paper can be accepted.